# New Bicyclic Pyridine-Based Hybrids Linked to the 1,2,3-Triazole Unit: Synthesis via Click Reaction and Evaluation of Neurotropic Activity and Molecular Docking

**DOI:** 10.3390/molecules28030921

**Published:** 2023-01-17

**Authors:** Samvel N. Sirakanyan, Domenico Spinelli, Anthi Petrou, Athina Geronikaki, Victor G. Kartsev, Elmira K. Hakobyan, Hasmik A. Yegoryan, Luca Zuppiroli, Riccardo Zuppiroli, Armen G. Ayvazyan, Ruzanna G. Paronikyan, Tatevik A. Arakelyan, Anush A. Hovakimyan

**Affiliations:** 1Scientific Technological Center of Organic and Pharmaceutical Chemistry of National Academy of Science of Republic of Armenia, Institute of Fine Organic Chemistry of A.L.Mnjoyan, Ave. Azatutyan 26, Yerevan 0014, Armenia; 2Dipartimento di Chimica G. Ciamician, Alma Mater Studiorum-Università di Bologna, Via F. Selmi 2, 40126 Bologna, Italy; 3School of Pharmacy, Aristotle University of Thessaloniki, 54124 Thessaloniki, Greece; 4InterBioScreen, 119019 Moscow, Russia; 5Department of Industrial Chemistry ‘Toso Montanari’, Alma Mater Studiorum-Università di Bologna, Viale del Risorgimento 4, 40136 Bologna, Italy; 6Scientific Technological Center of Organic and Pharmaceutical Chemistry of National Academy of Science of Republic of Armenia, Molecule Structure Research Centre, Ave. Azatutyan 26, Yerevan 0014, Armenia

**Keywords:** condensed pyridines, hybrids, click reaction, neurotropic activity

## Abstract

The synthesis of new original bicyclic pyridine-based hybrids linked to the 1,2,3-triazole unit was described via a click reaction. The anticonvulsant activity and some psychotropic properties of the new compounds were evaluated. The biological assays demonstrated that some of the studied compounds showed high anticonvulsant and psychotropic properties. The five most active compounds (**7a**, **d**, **g**, **j**, and **m**) contain a pyrano [3,4-*c*]pyridine cycle with a methyl group in the pyridine ring in their structures. Furthermore, molecular docking studies were performed, and their results are in agreement with experimental data.

## 1. Introduction

According to the WHO, more than 50 million people worldwide suffer from epilepsy [1]. Epilepsy is one of the most severe diseases of the nervous system, leading to serious complications and consequences in the form of reduced ability to work, social maladaptation, disability, and even premature death in patients [2]. As a result, the treatment of epilepsy appears to be a very important social and socio-economic problem. The primary tasks of the current stage of the treatment of epilepsy and epileptic syndromes are: the widespread introduction of new antiepileptic drugs with innovative mechanisms of action on the “target” of the pathological epileptic system; the use of new drugs not only as an additional therapy for drug-resistant epilepsy, but, perhaps, a faster transition to new forms of drugs at the earliest stages of ineffective treatment with basic drugs [3]. In recent years, with the use of antiepileptic drugs, mainly of the second generation, there has been a tendency to optimize results, aimed at the use of anticonvulsants with extended combined properties [4]. Hence, epileptic mood disorders, twilight clouding of consciousness, epileptic psychoses, epileptic personality changes, and emotional disturbances are known. A new treatment approach allows such drugs to be used in various related areas of neurology and psychiatry when it becomes necessary to alleviate emotional stress, induce sedation, stabilize or prevent mood swings in bipolar disorders of manic-depressive states [5].

Hybridization of bioactive natural and non-natural compounds is one of the most promising and fundamentally novel approaches for the design of new lead structures and the discovery of novel and potent drugs in medicinal chemistry [6,7,8,9,10,11,12,13]. According to biological tests, hybrid molecules, which are generally synthetic compounds, are built of two or more natural or synthetically derived fragments that are linked by at least one carbon-carbon bond and are, in many cases, more active than their respective parent compounds [6,7,8,9,10,11].

In our previous work we demonstrated that the terminal triple bond was obtained based on the 3(6)-hydroxy derivatives of bicyclic pyridines **1**, and the compound was found to have strong antiplatelet and vasodilatory potential, as well as an IC_50_ twice as low as that of the clinically used acetylsalicylic acid [14]. The azido group was synthesized based on the 6-piperazino derivatives of pyrano [3,4-*c*]pyridines, which show pronounced neurotropic [15,16] and antimicrobial [17] activities. On the other hand, the obtained hybrids contain a 1,2,3-triazole ring, whose derivative can show a wide spectrum of biological properties [18,19,20].

Taking into account the above-mentioned results, herein we describe the synthesis of new original hybrid compounds as potential biologically active substances while studying their neurotropic properties. The click chemistry [21,22,23,24] was used to synthesize the corresponding terminal triple bond and azido group based on the biologically active bicyclic pyridine derivatives.

## 2. Results and Discussion

### 2.1. Chemistry

We synthesized terminal alkines from the 3(6)-hydroxy derivatives of cyclopenta[*c*]pyridine **1a** [25], 5,6,7,8-tetrahydroisoquinolines **1b**–**d** [25], pyrano [3,4-*c*]pyridines **1e**–**g** [14,26], and 2,7-naphtyridines **1h**,**i** [27] to obtain the aimed hybrids, the 1,2,3-triazoles. It should be mentioned that tautomerism is possible in compound **1** lactam-lactim (NH/OH) via proton-migration of a hydrogen atom between the two basic centers. However, the spectroscopic data (IR, NMR spectra, and X-ray data) strongly evidenced that pyrano [3,4-*c*]pyridines **1** in the crystalline state and in solution exist only in the lactam (NH) tautomeric form [26]. Interestingly, in alkaline solution the situation changes, where by the alkylation of condensed pyridines **1** in basic conditions we obtained high yields of *O*-alkylated derivatives [24,28]. The same situation was observed during the reaction of compound **1** with propargyl bromide. As a result, relevant *O*-alkylated derivatives **2a**–**i** [24] were obtained under basic conditions at room temperature in high yields (Figure 1, Table 1).

However, in an accurate examination of the crude reaction mixture by ^1^H NMR spectroscopy, we have detected the traces of *N*-alkylated compounds **3** in ratios of 3–10% with the corresponding **2**, which completely disappeared after the recrystallization of the mixture from ethanol.

In the case of compound **3f**, we succeeded in isolating and identifying it in its pure state from the reaction mixture with the corresponding compound **2f**, due to its different solubility in the ethanol. The structures of compounds **2a**–**i** and **3f** were confirmed by IR, NMR, X-ray spectroscopy, and elemental analysis (see Appendix A).

Compounds **2f** and **3f** showed different physico-chemical data. For example, in the ^1^H NMR spectra, the doublet signal of the OCH_2_ group in **2f** was displayed at 5.06 ppm, while in isomeric **3f** the doublet signal of the NCH_2_ group was shifted up-field to 4.88 ppm. The remaining signals from the protons showed only slight differences. In the IR spectra of compound **3f**, the absorption band characteristic for the carbonyl group at 1633 cm^–1^ was also observed, while for compounds **2a**–**i** it was absent. It was also interesting to compare the X-ray data of compounds **2f** and **3f**. Crystallographic and experimental data for isomer compounds **2f** and **3f** are listed in Table 2.

The molecular structure of the compound **3f** is shown in Figure 1. Conformational calculations of cyclic fragments showed that the pyridine ring is perfectly planar, with the maximum deviations of atoms from the mean-squared plane not exceeding 0.0284(1) Å. The tetrahydro-1*H*-pyran ring shows well-expressed “half-chair” conformation. The four atoms (C5, C6, C19, and C20) of the ring lie in the same plane (the maximal shift from plane was 0.0202(1) Å) while atoms O1 and C2 are out of plane with shifts from “half-chair” plane of −0.3172(1) Å and 0.4097(1) Å, respectively. In 3D packing, the intermolecular interactions could be described mainly via van der Waals forces.

The structure of **2f** depicted in Figure 1 appears to be similar to that of **3f**, with the difference that the prop-1-yne group is linked to an oxygen (O11) atom at the pyridine ring (Figure 1). All bond lengths and angles in the molecules are in good agreement with each other and with their average statistical values. In this structure, the pyridine ring additionally has a planar conformation, with the maximum deviation of atoms from the mean-squared plane not exceeding 0.0144(4) Å. The tetrahydro-1*H*-pyran ring also shows “half-chair” conformation. The atoms C5, C6, C19, and C20 are located in the same plane (the maximal shift from plane was 0.0079(4) Å). In contrast atoms O1 and C2 are out of the “half-chair” plane showing maximal shifts of 0.3044(4) Å and −0.4616(4)Å, respectively. In 3D packaging, intermolecular interactions are mainly due to van der Waals forces.

In order to obtain the second component, namely azides, we started with 6-piperazino derivatives of pyrano [3,4-c]pyridines **4a**–**c** [15]. Thus, by acylation of compound **4** with chloroacetyl chloride, the relevant 1(8)-alkyl-3(6)-[4-(chloroacetyl)piperazin-1-yl]pyridine-4(5)-carbonitriles **5a**–**c** [24] were obtained. Further reaction of compound **5** with sodium azide in acetone led to the formation of 1(8)-alkyl-3(6)-[4-(azidoacetyl)piperazin-1-yl][*c*]pyridine-4(5)-carbonitriles **6a**–**c** [24] in high yields (Figure 2).

The synthesis of new bioactive hybrids was carried out according to the proposed general Figure 3 via reactions between the propargylated compounds **2** and azido derivatives **6** under the Cu-catalyzed azide-alkyne click reaction [24]. In the majority of the reactions, the hybrids **7a**–**u** were obtained in high yields (Y = 72–87%; Table 3). Their structures were confirmed by IR, NMR, MS spectroscopy, and elemental analysis (see Appendix A).

In the ^1^H NMR spectra of compounds **7a**–**u** the protons of the CH group of the 1,2,3-triazole ring were observed at 7.93–8.01 ppm, while the signals of C≡CH group were absent. The ^13^C NMR and MS spectra also confirmed their structure.

### 2.2. Biological Evaluation

The study of the neurotropic activity of 21 newly synthesized heterocyclic compounds (hybrids **7a**–**u**) was carried out according to indicators characterizing anticonvulsant, sedative, and anti-anxiety activity and side effects.

The study of the anticonvulsant activity of hybrids **7a**–**u** was assessed by evaluating the antagonism to pentylenetetrazole (PTZ), thiosemicarbazide convulsive seizures, and maximal electroshock-induced seizures (MES) [29,30,31,32,33]. The PTZ-induced test is considered an experimental model for the clonic component of epilepsy seizures and the prognostic anxiolytic activity of the compounds. PTZ is a common convulsive agent used in animal models to investigate the mechanisms of seizures. The model of thiosemicarbazide (TSC) seizures (affecting the exchange of GABA) causes clonic seizures similar to pentylenetetrazole ones. The MES test is used as an animal model for the generalization of tonic seizures in epilepsy [34,35]. The neurotoxicity and side effects of the compounds (movement coordination disorder, myorelaxation, and ataxia) were also studied on mice with the “rotating rod”[29] test.

To determine the 50% effective (ED_50_, causing the anticonvulsant effect in 50% of animals, calculated by the test of antagonism to PTZ), 50% neurotoxic (TD_50_, causing myorelaxation effect in 50% of animals, calculated by the “rotating rod” test), and 50% lethal (LD_50_, causing death in 50% of animals) doses, a statistical method of probit analysis proposed by Litchfield and Wilcoxon [36] was used. From a practical point of view, the active compounds’ therapeutic (TI = LD_50_/ED_50_) indexes were identified. The well-known antiepileptic drug ethosuximide and the tranquilizer diazepam were used as reference drugs [37].

The evaluation of the anticonvulsant activity of all the synthesized hybrid compounds revealed that they, to varying degrees, exhibit pentylentetrazole antagonism. The anti-PTZ activity of the investigated compounds ranges from 20 to 80%. It was found that the studied compounds are superior to ethosuximide in anticonvulsant activity by antagonism to pentylenetetrazole but inferior to diazepam (Table 4, Figure 2). However, unlike diazepam, they do not induce muscle relaxation in the studied doses 50, 200, and 500 mg/kg. They are low-toxic; their acute daily toxicity is in the range of over 650–930 mg/kg, and their therapeutic (TI) indexes are much greater than those of the reference drug ethosuximide (Table 4, Figure 2). Among all, compounds **7a**, **d**, **g**, **j**, **m**, and **o** have the highest therapeutic indexes. The order of activity of tested compounds toward the PTZ test can be presented as follows: **7m** > **7d** > **7g** > **7a** > **7j** > **7r** = **7n** > **7h** > **7o** = **7f** > **7t** > **7u** > **7i** > **7c** > **7q**. The first five most active hybrids in their structure contain a pyrano [3,4-*c*]pyridine cycle with a methyl group in the 8 position of the pyridine ring.

Further, the 15 most active hybrids (**7a**, **c**, **d**, **f**–**j**, **m**–**o**, **q**, **r**, **t**, and **u**) were selected to be studied on the model of thiosemicarbazide (TSC) seizures (Table 4). The compounds at the dose of 100 mg/kg increased the latency of thiosemicarbazide seizures by 1.5–4.37 times compared to the control. Ethosuximide has approximately the same effect as increasing the latent time of TSC convulsions. Diazepam is not effective in this model. The most active compounds in this test appeared to be **7u** and **7t**, while the least active was **7h**.

The data on the anticonvulsant activity of 15 active compounds in the MES test in comparison with the reference drugs ethosuximide and diazepam in mice are presented in Table 4. According to the MES test, the compounds studied as well as the reference drugs did not exhibit an anticonvulsant effect since they did not protect from tonic and clonic seizures caused by MES.

Next, these compounds (**7a**, **c**, **d**, **f**–**j**, **m**–**o**, **q**, **r**, **t**, and **u**) were studied on the “open field” [38,39], “elevated plus maze” (EPM) [40,41], “forced swimming” [42] tests at a dose of 50 mg/kg, since the ED_50_ of these compounds is within 50 mg/kg at the confidence intervals. For the most active compounds, the conditioned response of passive avoidance (CRPA) test was used to identify learning and memory processes [29,43]. This trend is due to the fact that many of the well-known anticonvulsants, while causing a therapeutic effect, grossly disrupt thought processes, reduce the intellectual abilities of a person, make memorization difficult, and cause amnesia.

The results of the activity of compounds and reference drugs in the “open field” model in rats are shown in Table 5 and Figure 3. As can be seen from the table, the studied compounds statistically significantly increase the number of horizontal and, in some cases, vertical movements. This indicates the activating effect of the compounds. Diazepam showed an activating effect too, in contrast to ethosuximide, which has neither an activating nor a sedative effect at the studied dose. Most of the considered compounds, like diazepam, increase the number of cells examined. The data obtained indicate the anxiolytic (anti-anxiety) activity of the compounds, which is especially pronounced in compounds **7a**, **g**, **m**, **j**, and **o**. Diazepam at a dose of 2 mg/kg has the same properties, in contrast to ethosuximide. The order of activity of tested compounds in this model is: **7m** > **7g** > **7j** > **7a** = **7o**. The most active compounds were the top five of the previous test. It should be noted that compound **7o**, containing two pyrano [3,4-*c*]pyridine rings, has almost the same activity as that of compound **7a**.

In the elevated plus-maze (EPM) model, control animals are predominantly located in closed arms (Table 6, Figure 4). All compounds as well as ethosuximide and diazepam increased, in a statistically significant manner, the time spent by experimental animals in the center. Statistically, the investigated compounds significantly reduce the residence time in the closed arms and the number of entries into the closed arms. After the administration of all compounds, the experimental animals, in contrast to the control animals and those who received ethosuximide at a dose of 200 mg/kg, enter the open arms and stay there for 6.6 (compound **7g**) to 66.4 (compound **7m**) sec. The administration of these compounds, as well as diazepam, in fact leads to the identification of an anxiolytic effect. In this test again, the most active compound was found to be **7m**, followed by **7i**.

The forced swimming test (FST) is used to monitor depressive-like behavior and is based on the assumption that immobility reflects a measure of behavioral despair. On the “forced swimming” model, some of the selected compounds (**7a**, **c**, **g**, **h**, **j**, **m**, **o**, and **q**) increase the active swimming time and the latent period of the first immobilization (Table 7, Figure 5). In control mice, the first immobilization occurs after 52 s. This indicates that the studied compounds at a dose of 50 mg/kg exhibit some antidepressant effect. The data regarding ethosuximide at a dose of 200 mg/kg are comparable with the control data. The reference drug diazepam at a dose of 2 mg/kg acts similarly to the noted compounds, increasing the latent time of the first immobilization. At the same time, diazepam significantly reduces the total immobilization time, while the compounds **7a**, **c**, **g**, **h**, **j**, **m**, **o**, **q**, and ethosuximide increase this time.

On the model of electroshock retrograde amnesia—conditioned response of passive avoidance (CRPA) (Table 8) within 6 min, control rats on the 1st and 2nd days of the experiment remained in the light compartment for almost the entire period (280.0 s). Under conditions of retrograde amnesia in rats, the administration of compounds at a dose of 50 mg/kg a day, except for compound **7j**, causes some increase in the time of reproduction of the reflex in animals. Statistically, these values are significantly different from the control values and indicate the antiamnesic effect of the compounds. Similar data were obtained for the nootropic drug piracetam, in contrast to diazepam, which did not have an antiamnestic effect (diazepam under the same conditions even leads to a decrease in behavioral parameters compared to the control, as well as the compound **7j**) (Table 8, Figure 6).

### 2.3. Molecular Docking

#### 2.3.1. Docking Studies for Prediction of the Mechanism of Anticonvulsant and Anxiolytic Activity (Docking to GABA_A_ Receptor)

It is widely known that antiepileptic drugs target GABA_A_ receptors in order to block sodium channels or enhance γ-aminobutyric acid (GABA) function [44,45]. Therefore, docking studies of all tested compounds were performed in order to get a better understanding of the GABA_A_ receptor inhibitory potency at the molecular level and shed light on the interactions in the active site of the GABA_A_ receptor.

For docking studies, the crystal structure of the GABA_A_ receptor was retrieved from the protein data bank (PDB) with PDB ID: 4COF obtained via X-ray method [46]. The X-ray diffraction structure of the GABA_A_ receptor had a resolution of 2.97 Å, an R value of 0.206, and an R free value of 0.226. As a first step in docking studies and for the validation of docking parameters, the initial co-crystal ligand benzamidine was extracted and re-docked at the catalytic site of a protein using the same parameters and preparation steps as used for the tested compounds. The root-mean-square deviation (RMSD) between co-crystal and re-docked poses was found to be 0.34 Å (Figure 7).

Docking results presented in Table 9 revealed that compound **7j** bound tightly in the active site of the GABA_A_ receptor with a value of free binding energy of −10.36 kcal/mol, forming three hydrogen bonds between the nitrogen atoms and residues Tyr97 (N···H, 3.31Å), Thr202 (N···H, 3.13Å) and Tyr205 (N···H, 2.95Å), while diazepam forms only one hydrogen bond with Thr202 (N···H, 2.67Å, Figure 8 and Figure 9). Furthermore, the furan moiety of the compound showed hydrophobic interactions with the residues Ala201 and Phe200, further stabilizing the complex ligand-enzyme.

#### 2.3.2. Docking to the SERT Transporter and the 5-HT_1A_ Receptor

Antidepressant drugs fall into two main categories: tricyclic antidepressants and selective serotonin reuptake inhibitors (SSRIs) [47]. These drugs inhibit the transport of serotonin into the pre-synaptic neuron by inhibiting the serotonin (5-HT) transporter (SERT). SERT is a trans-membrane protein located in the membrane of pre-synaptic neurons that removes serotonin from the synaptic cleft, resulting in the termination of serotonergic neurotransmission. The increase in serotonin activates the 5-HT_1A_ receptors, decreasing the serotonergic neurotransmission and causing a delay in the onset of antidepressant action [48,49]. This delay lasts until HT_1A_ receptors become desensitized and the release of serotonin is normalized.

Taking all the above into account, in order to study if the tested compounds act as dual inhibitors of serotonin transporter (SERT) and alongside antagonize the pre-synaptic auto inhibitory 5-HT_1A_ receptors, we proceeded with docking studies in SERT transporter and 5-HT_1A_ receptor.

As there is no available crystal structure of the SERT transporter in the protein data nank (PDB) we used the x-ray crystal structure of LeuT bound to L-Tryptophan (PDB code: 3F3A), a prokaryotic homologue of SERT [50]. The results of docking studies on the SERT transporter are presented in Table 10.

As a first step in docking studies and for the validation of docking parameters, the initial co-crystal ligand L-Tryptophan was extracted and re-docked at the binding site of the protein using the same parameters and preparation steps as used for the tested compounds. The RMSD between co-crystal and re-docked poses was found to be 0.86 Å (Figure 10).

Compound **7a** showed the best docking score, forming two hydrogen bonds. The first one was between the nitrogen atom of the CN group and the hydrogen atom of the side chain of Lys443 (distance 2.42 Å). The second was between the nitrogen atom of the thiazole ring of the compound and the hydrogen of the side chain of Arg11 (distance 2.53 Å) (Figure 11). The docking results coincide with the experimental data.

For docking to the 5-HT_1A_ receptor, the crystal structure of the human β2-adrenergic receptor in complex with the neutral antagonist alprenolol (PDB code: 3NYA) was used [51,52]. For the validation of docking parameters, the initial co-crystal ligand alprenolol was re-docked at the catalytic site of protein, and the root-mean-square deviation (RMSD) between the co-crystal and the re-docked pose was found to be 0.98 Å (Figure 12).

All the tested compounds were docked into the orthosteric binding site of the 5-HT_1A_ receptor (Table 11), and the best docking score was found for compound **7a** (−11.23 kcal/mol), which formed five hydrogen bonds with the residues Tyr118, Tyr199, Ser204, Asn312, and Tyr316. Additionally, hydrophobic interactions were observed between methyl substituents of the compound and residues Trp109, Ile309, Tyr308, Phe193, Val114, Tyr110, Ile201, Val297, and Ala200 (Figure 13A). It is worth noting that alprenolol forms hydrogen bonds with the same Tyr316 and Asn312 residues as compound **7a** does. Moreover, this compound is oriented in the same cavity of the enzyme as alprenolol (Figure 13B). This may justify the high activity of compound **7a**. Finally, the docking studies revealed that this compound can probably be a dual target molecule since it seems to be a good inhibitor of the SERT transporter and a good 5-HT_1A_ receptor binder.

### 2.4. Drug Likeness

All tested compounds were evaluated for their absorption, distribution profile, and drug-likeness model score (a combined effect of physico-chemical properties, pharmacokinetics, and pharmacodynamics of a compound represented by a numerical value) was computed by MolSoft, (MolSoft, 2007)) and the results are presented in Table 12. As depicted in Figure 9, those with green coloring indicate non drug-like behavior, while those with blue coloring are considered drug-like. Compounds having a zero or negative value cannot be considered drug-like. The drug-likeness score was found to be from 0.33 to 0.83 for all compounds, thus they can be treated as drug candidates (Figure 14).

The ADMET structure-activity relationship server, admetSAR [2], used for this prediction, gives a probability for each ADMET descriptor. More than 40 high predictive models were implemented in admetSAR, which were trained by state-of-the-art machine learning methods. According to prediction, all the tested molecules were able to pass the blood-brain barrier (BBB). We further looked into the possibility of our compounds being P-glycoprotein substrates or inhibitors, which play a critical role in pharmacokinetic properties, particularly distribution and excretion (Table 12). The admetSAR server revealed that all compounds are both good P-glycoprotein substrates (65–73%) and inhibitors (74–78%).

## 3. Materials and Methods

### 3.1. Chemistry

#### 3.1.1. General Information

^1^H and ^13^C NMR spectra were recorded in DMSO-*d_6_*/CCl_4_ (1/3) solution (300 MHz for ^1^H and 75 MHz for ^13^C, respectively) on a Mercury 300VX spectrometer (Varian Inc., Palo Alto, CA, USA). Chemical shifts were reported as *δ* (parts per million) relative to TMS as internal standard. The IR spectra were recorded on a Nicolet Avatar 330-FT-IR spectrophotometer (Thermo Nicolet, CA, USA) in vaseline, *ν_max_* in cm^–1^. MS spectra were recorded on the Waters Xevo Q-Tof. Melting points were determined on a MP450 melting point apparatus. Elemental analyses were performed on an Elemental Analyzer Euro EA 3000. Crystallographic and experimental data are listed in Table 2. Deposition number 2201666 (for compound **2f**) and 1937036 (for compound **3f**) contain the supplementary crystallographic data for this paper. The full crystallographic data in CIF format is available free of charge via the internet at: www.ccdc.cam.ac.uk/structures (accessed on 22 December 2022). The unit cell parameters of the crystals of the compounds **2f** and **3f** were measured on an Enraf-Nonius automated diffractometer CAD-4 at room temperature using the diffraction angles of 24 reflections. The diffraction experiment was performed on the same diffractometer using graphite monochromated MoKα-radiation and a θ/2θ-scan measurement method. The structures were solved by the direct method and refined using the software package SHELXTL [53]. All non-hydrogen atoms were refined in anisotropic approximation by full-matrix least squares methods. The hydrogen atoms were positioned geometrically and refined using a riding model. Compounds **2a**, **c**, **e** [24]; **5b**, **c** [24]; **6b**, **c** [24]; and **7b**, **h**, **o** [24] were already described.

#### 3.1.2. General Procedure for the Synthesis of Propargylated Derivatives **2b**, **d**, **f**, **g**–**I**, and **3f**

To a stirred suspension of compound **1** (5 mmol) and potassium carbonate (1.4 g, 10 mmol) in absolute DMF (25 mL), the propargyl bromide (0.42 mL, 5.5 mmol) was added in a dropwise manner. The reaction mixture was maintained at room temperature for 5 h. After water was added (50 mL), the resulting crystals were filtered off, washed with water, dried, and recrystallized from ethanol (for compound **3f**, chloroform was used).

*1-Methyl-3-(prop-2-yn-1-yloxy)-5,6,7,8-tetrahydroisoquinoline-4-carbonitrile* (**2b**). Colorless solid; yield 71%; mp 94–96 °C; IR *ν*/cm^–1^: 2128 (C≡CH), 2221 (C≡N), and 3246 (≡CH). ^1^H NMR (300 MHz, DMSO-*d_6_*/CCl_4_, 1/3): δ 1.76–1.89 (m, 4H, 6,7-CH_2_), 2.41 (s, 3H, CH_3_), 2.57–2.62 (m, 2H, 8-CH_2_), 2.84–2.89 (m, 2H, 5-CH_2_), 2.90 (t, *J =* 2.4 Hz, 1H, ≡CH), and 5.02 (d, *J =* 2.4 Hz, 2H, OCH_2_). ^13^C NMR (75 MHz, DMSO-*d_6_*/CCl_4_, 1/3): δ 20.84, 21.80, 21.85, 24.62, 27.82, 53.09, 75.80, 78.01, 92.92, 113.27, 124.16, 152.01, 158.28, and 159.31. Anal. calcd. for C_14_H_14_N_2_O: C 74.31; H 6.24; and N 12.38%. Found: C 74.69; H 6.44; and N 12.64%.

*1-(2-Furyl)-3-(prop-2-yn-1-yloxy)-5,6,7,8-tetrahydroisoquinoline-4-carbonitrile* (**2d**). Cream solid; yield 74%; mp 162–164 °C; IR *ν*/cm^–1^: 2121 (C≡CH), 2217 (C≡N), and 3291 (≡CH). ^1^H NMR (300 MHz, DMSO-*d_6_*/CCl_4_, 1/3): δ 1.80–1.90 (m, 4H, 6,7-CH_2_), 2.91–2.97 (m, 2H, 8-CH_2_), 2.93 (t, *J =* 2.4 Hz, 1H, ≡CH), 3.01–3.07 (m, 2H, 5-CH_2_), 5.08 (d, *J =* 2.4 Hz, 2H, OCH_2_), 6.59 (dd, *J =* 3.4, 1.7 Hz, 1H, 4-CH_fur._), 7.19 (dd, *J =* 3.4, 0.7 Hz, 1H, 3-CH_fur._), and 7.70 (dd, *J =* 1.7, 0.7 Hz, 1H, 5-CH_fur._). ^13^C NMR (75 MHz, DMSO-*d_6_*/CCl_4_, 1/3): δ 20.59, 21.89, 25.40, 28.48, 53.35, 75.87, 77.93, 93.69, 111.49, 113.24, 114.25, 122.42, 144.10, 146.35, 152.82, 154.23, and 158.87. Anal. calcd. for C_17_H_14_N_2_O_2_: C 73.37; H 5.07; and N 10.07%. Found: C 73.70; H 5.24; and N 10.29%.

*8-Ethyl-3,3-dimethyl-6-(prop-2-yn-1-yloxy)-3,4-dihydro-1H-pyrano [3,4-c]pyridine-5-carbonitrile* (**2f**). Colorless solid; yield 70%; mp 146–148 °C; IR *ν*/cm^–1^: 2191 (C≡CH), 2222 (C≡N), andand 3267 (≡CH). ^1^H NMR (300 MHz, DMSO-*d_6_*/CCl_4_, 1/3): δ 1.28 (s, 6H, C(CH_3_)_2_), 1.28 (t, *J =* 7.4 Hz, 3H, CH_2_CH_3_), 2.62 (q, *J =* 7.4 Hz, 2H, CH_2_CH_3_), 2.77 (s, 2H, CH_2_), 2.94 (t, *J =* 2.4 Hz, 1H, ≡CH), 4.63 (s, 2H, OCH_2_), 5.06 (d, *J =* 2.4 Hz, 2H, OCH_2_). ^13^C NMR (75 MHz, DMSO-*d_6_*/CCl_4_, 1/3): δ 10.82, 25.66, 26.23, 37.70, 53.32, 58.78, 68.93, 75.94, 77.90, 92.99, 112.93, 120.80, 148.94, 159.32, and 160.18. Anal. calcd. for C_16_H_18_N_2_O_2_: C 71.09; H 6.71; and N 10.36%. Found: C 71.44; H 6.90; and N 10.61%.

*3,3-Dimethyl-8-propyl-6-(prop-2-yn-1-yloxy)-3,4-dihydro-1H-pyrano [3,4-c]pyridine-5-carbonitrile* (**2g**). Colorless solid; yield 76%; mp 152–154 °C; IR *ν*/cm^–1^: 2136 (C≡CH), 2220 (C≡N), and 3281 (≡CH). ^1^H NMR (300 MHz, DMSO-*d_6_*/CCl_4_, 1/3): δ 1.00 (t, *J =* 7.4 Hz, 3H, CH_2_CH_3_), 1.27 (s, 6H, C(CH_3_)_2_), 1.71–1.84 (m, 2H, CH_2_CH_3_), 2.56 (t, *J =* 7.4 Hz, 2H, CH_2_C_2_H_5_), 2.77 (s, 2H, CH_2_), 2.93 (t, *J =* 2.4 Hz, 1H, ≡CH), 4.63 (s, 2H, OCH_2_), and 5.05 (d, *J =* 2.4 Hz, 2H, OCH_2_). ^13^C NMR (75 MHz, DMSO-*d_6_*/CCl_4_, 1/3): δ 13.44, 19.97, 25.66, 34.91, 37.72, 53.32, 58.90, 68.93, 75.93, 77.88, 93.09, 112.93, 121.12, 149.04, 158.44, and 160.06. Anal. calcd. for C_17_H_20_N_2_O_2_: C 71.81; H 7.09; and N 9.85%. Found: C 72.12; H 7.24; and N 10.06%.

*1-Azepan-1-yl-7-isopropyl-3-(prop-2-yn-1-yloxy)-5,6,7,8-tetrahydro-2,7-naphthyridine-4-carbonitrile* (**2h**). Yellow solid; yield 72%; mp 95–97 °C; IR *ν*/cm^–1^: 2127 (C≡CH), 2207 (C≡N), and 3245 (≡CH). ^1^H NMR (300 MHz, DMSO-*d_6_*/CCl_4_, 1/3): δ 1.08 (d, *J =* 6.5 Hz, 6H, CH(CH_3_)_2_), 1.56–1.64 (m, 4H, C_6_H_12_N), 1.79–1.88 (m, 4H, C_6_H_12_N), 2.67–2.73 (m, 2H, NCH_2_CH_2_), 2.81–2.90 (m, 4H, NCH_2_CH_2_, ≡CH, CH(CH_3_)_2_), 3.40 (s, 2H, NCH_2_), 3.57–3.63 (m, 4H, N(CH_2_)_2_), and 4.93 (d, *J =* 2.4 Hz, 2H, OCH_2_). ^13^C NMR (75 MHz, DMSO-*d_6_*/CCl_4_, 1/3): δ 18.09, 26.24, 28.20, 29.13, 44.22, 49.51, 50.82, 52.68, 52.74, 52.80, 53.32, 75.39, 78.46, 82.95, 114.60, 150.66, 158.39, and 159.21. Anal. calcd. for C_21_H_28_N_4_O: C 71.56; H 8.01; and N 15.90%. Found: C 71.92; H 8.20; and N 16.15%.

*1-Azepan-1-yl-7-benzyl-3-(prop-2-yn-1-yloxy)-5,6,7,8-tetrahydro-2,7-naphthyridine-4-carbonitrile* (**2i**). Yellow solid; yield 73%; mp 46–48 °C; IR *ν*/cm^–1^: 2125 (C≡CH), 2206 (C≡N), and 3248 (≡CH). ^1^H NMR (300 MHz, DMSO-*d_6_*/CCl_4_, 1/3): δ 1.47–1.56 (m, 4H, C_6_H_12_N), 1.69–1.78 (m, 4H, C_6_H_12_N), 2.67–2.73 (m, 2H, NCH_2_CH_2_), 2.85–2.91 (m, 3H, NCH_2_CH_2_, ≡CH), 3.34 (s, 2H, NCH_2_), 3.51–3.57 (m, 4H, N(CH_2_)_2_), 3.65 (s, 2H, CH_2_Ph), 4.93 (d, *J =* 2.4 Hz, 2H, OCH_2_), and 7.18–7.30 (m, 5H, Ph). ^13^C NMR (75 MHz, DMSO-*d_6_*/CCl_4_, 1/3): δ 18.12, 26.10, 28.06, 28.56, 48.35, 50.73, 52.77, 53.78, 61.75, 75.38, 75.41, 82.81, 110.25, 114.58, 126.58, 127.68, 128.20, 137.42, 150.27, 158.16, and 159.31. Anal. calcd. for C_25_H_28_N_4_O: C 74.97; H 7.05; and N 13.99%. Found: C 75.29; H 7.21; and N 14.22%.

*8-Ethyl-3,3-dimethyl-6-oxo-7-(prop-2-yn-1-yloxy)-3,4,6,7-tetrahydro-1H-pyrano [3,4-c]pyridine-5-carbonitrile* (**3f**). Colorless solid; yield 10%; mp 163–165 °C; IR *ν*/cm^–1^: 1633 (C=O), 2131 (C≡CH), 2223 (C≡N), and 3256 (≡CH). ^1^H NMR (300 MHz, DMSO-*d*_6_/CCl_4_, 1/3): δ 1.28 (s, 6H, C(CH_3_)_2_), 1.28 (t, *J =* 7.5 Hz, 3H, CH_2_CH_3_), 2.71 (s, 2H, CH_2_), 2.77 (q, *J =* 7.5 Hz, 2H, CH_2_CH_3_), 2.89 (t, *J =* 2.4 Hz, 1H, ≡CH), 4.56 (s, 2H, OCH_2_), and 4.88 (d, *J =* 2.5 Hz, 2H, NCH_2_). ^13^C NMR (75 MHz, DMSO-*d_6_*/CCl_4_, 1/3): δ 11.56, 21.87, 25.89, 32.79, 38.38, 58.36, 69.20, 73.89, 77.72, 99.91, 110.22, 114.22, 151.33, 154.74, and 158.28. Anal. calcd. for C_16_H_18_N_2_O_2_: C 71.09; H 6.71; and N 10.36%. Found: C 71.46; H 6.88; and N 10.62%.

#### 3.1.3. Procedure for the Synthesis of 6-[4-(Chloroacetyl)piperazin-1-yl]-3,3,8-trimethyl-3,4-dihydro-1*H*-pyrano [3,4-*c*]pyridine-5-carbonitrile (**5a**)

To a stirred solution of compound **4a** (1.43 g, 5 mmol) and trimethylamine (0.84 mL, 6 mmol) in benzene (50 mL) the chloroacetyl chloride (4.8 mL, 6 mmol) was added in a dropwise manner. The reaction mixture was maintained at 35 °C for 6 h. Then the reaction mixture was cooled at room temperature, the solvent was removed under vacuum and water (50 mL) was added. The resulting crystals were filtered off, washed with water, dried, and recrystallized from ethanol. Colorless solid; yield 88%; mp 129–130 °C; IR ν/cm^–1^: 1666 (C=O) and 2207 (C≡N). ^1^H NMR (300 MHz, DMSO-d_6_/CCl_4_, 1/3): δ 1.27 (s, 6H, C(CH_3_)_2_), 2.30 (s, 3H, CH_3_), 2.70 (s, 2H, CH_2_), 3.54–3.71 (m, 8H, C_4_H_8_N_2_), 4.23 (s, 2H, CH_2_Cl), and 4.56 (s, 2H, OCH_2_). ^13^C NMR (75 MHz, DMSO-d_6_/CCl_4_, 1/3): δ 20.97, 25.73, 37.72, 40.80, 41.16, 45.11, 47.56, 48.11, 59.19, 69.04, 93.44, 115.47, 119.52, 147.75, 155.35, 159.11, and 164.23. Anal. calcd. For C_18_H_23_ClN_4_O_2_: C 59.58; H 6.39; and N 15.44%. Found: C 59.92; H 6.55; and N 15.68%.

#### 3.1.4. Procedure for the Synthesis of 6-[4-(Azidoacetyl)piperazin-1-yl]-3,3,8-trimethyl-3,4-dihydro-1*H*-pyrano [3,4-*c*]pyridine-5-carbonitrile (**6a**)

A mixture of compound **5a** (1.81 g, 5 mmol) and sodium azide (0.65 g, 10 mmol) in acetone (50 mL) was heated at reflux for 15 h. After filtration the solvent was evaporated, and water (50 mL) was added. The resulting crystals were filtered off, washed with water, dried, and recrystallized from ethanol. Colorless solid; yield 86%; mp 130–132 °C; IR ν/cm^−1^: 1650 (C=O), 2105 (N_3_), and 2211 (C≡N). ^1^H NMR (300 MHz, DMSO-*d_6_*/CCl_4_, 1/3): δ 1.27 (s, 6H, C(CH_3_)_2_), 2.29 (s, 3H, CH_3_), 2.70 (s, 2H, CH_2_), 3.48–3.72 (m, 8H, C_4_H_8_N_2_), 4.05 (s, 2H, CH_2_N_3_), and 4.56 (s, 2H, OCH_2_). ^13^C NMR (75 MHz, DMSO-d_6_/CCl_4_, 1/3): δ 20.99, 25.75, 37.74, 41.04, 43.84, 47.60, 47.97, 49.58, 59.20, 69.06, 93.33, 115.53, 119.48, 147.78, 155.40, 159.11, and 165.36. Anal. calcd. for C_18_H_23_N_7_O_2_: C 58.52; H 6.28; and N 26.54%. Found: C 58.91; H 6.48; and N 26.81%.

#### 3.1.5. General Procedure for the Synthesis of 1,2,3-Triazoles **7a**, **c**–**g**, **i**–**n**, **p**–**u**

Propargyl derivatives of fused pyridines **2** (2 mmol) and corresponding azides **6** (2.2 mmol) were suspended in a 1:1 mixture of water and *tert*-butyl alcohol (30 mL). Sodium ascorbate (0.04 g, 0.2 mmol, of freshly prepared solution in water) was added, followed by copper(II) sulfate pentahydrate (0.05 g, 0.2 mmol, in water). The mixture was stirred for 10 h at room temperature and then 10 h at 40–50 °C. After cooling water was added (50 mL), the precipitate was filtered off, washed with water, and recrystallized from a mixture of ethanol-chloroforme (1:3).

*6-{4-[(4-{[(4-Cyano-1-isopropyl-6,7-dihydro-5H-cyclopenta[c]pyridin-3-yl)oxy]methyl}-1H-1,2,3-triazol-1-yl)acetyl]piperazin-1-yl}-3,3,8-trimethyl-3,4-dihydro-1H-pyrano [3,4-c]pyridine-5-carbonitrile* (**7a**). Colorless solid; yield 78%; mp 221–223 °C; IR *ν*/cm^–1^: 1665 (C=O) and 2216 (C≡N). ^1^H NMR (300 MHz, DMSO-*d_6_*/CCl_4_, 1/3): δ 1.27 (d, *J =* 6.6 Hz, 6H, CH(CH_3_)_2_), 1.28 (s, 6H, C(CH_3_)_2_), 2.13–2.24 (m, 2H, 6’-CH_2_), 2.31 (s, 3H, CH_3_), 2.71 (s, 2H, 4-CH_2_), 2.90 (t, *J =* 7.4 Hz, 2H, 7′-CH_2_), 3.03 (t, *J =* 7.6 Hz, 2H, 5’-CH_2_), 3.05 (sp, *J =* 6.6 Hz, 1H, CH(CH_3_)_2_), 3.58–3.75 (m, 8H, C_4_H_8_N_2_), 4.57 (s, 2H, 1-CH_2_), 5.48 (s, 2H, NCH_2_), 5.55 (s, 2H, OCH_2_), and 7.94 (s, 1H, CH_triazole_). ^13^C NMR (75 MHz, DMSO-*d_6_*/CCl_4_, 1/3): δ 20.60, 20.99, 23.89, 25.73, 28.68, 32.00, 32.70, 37.73, 41.21, 43.97, 47.49, 47.94, 50.35, 59.19, 59.64, 69.04, 89.74, 93.24, 113.81, 115.55, 119.42, 125.19, 129.59, 141.79, 147.74, 155.39, 159.08, 161.01, 161.22, 162.61, and 163.78. Anal. calcd. for C_33_H_39_N_9_O_3_: C 65.01; H 6.45; and N 20.68%. Found: C 65.33; H 6.62; and N 20.91%. ESI + MS: [C_33_H_39_N_9_O_3_Na_1_]^+^ Found: 632.30747.

*6-{4-[(4-{[(4-Cyano-1-isopropyl-6,7-dihydro-5H-cyclopenta[c]pyridin-3-yl)oxy]methyl}-1H-1,2,3-triazol-1-yl)acetyl]piperazin-1-yl}-3,3-dimethyl-8-propyl-3,4-dihydro-1H-pyrano [3,4-c]pyridine-5-carbonitrile* (**7c**). Colorless solid; yield 73%; mp 240–241 °C; IR *ν*/cm^–1^: 1656 (C=O) and 2216 (C≡N). ^1^H NMR (300 MHz, DMSO-*d_6_*/CCl_4_, 1/3): δ 1.00 (t, *J =* 7.4 Hz, 3H, CH_2_CH_3_), 1.27 (d, *J =* 6.7 Hz, 6H, CH(CH_3_)_2_), 1.28 (s, 6H, C(CH_3_)_2_), 1.67–1.80 (m, 2H, CH_2_CH_3_), 2.13–2.24 (m, 2H, 6′-CH_2_), 2.48–2.54 (m, 2H, CH_2_C_2_H_5_), 2.72 (s, 2H, 4-CH_2_), 2.90 (t, *J =* 7.3 Hz, 2H, 7′-CH_2_), 3.03 (t, *J =* 7.6 Hz, 2H, 5´-CH_2_), 3.05 (sp, *J =* 6.7 Hz, 1H, CH(CH_3_)_2_), 3.59–3.76 (m, 8H, C_4_H_8_N_2_), 4.60 (s, 2H, 1-CH_2_), 5.48 (s, 2H, NCH_2_), 5.55 (s, 2H, OCH_2_), and 7.93 (s, 1H, CH_triazole_). ^13^C NMR (75 MHz, DMSO-*d_6_*/CCl_4_, 1/3): δ 13.53, 20.01, 20.61, 23.90, 25.74, 28.68, 32.01, 32.70, 35.20, 37.88, 41.20, 43.96, 47.40, 47.97, 50.34, 58.94, 59.66, 68.97, 89.76, 93.18, 113.82, 115.57, 119.08, 125.18, 129.58, 141.79, 147.96, 158.54, 159.06, 161.01, 161.23, 162.61, and 163.77. Anal. calcd. for C_35_H_43_N_9_O_3_: C 65.91; H 6.80; and N 19.77%. Found: C 66.31; H 7.01; and N 20.05%. ESI + MS: [C_35_H_43_N_9_O_3_Na_1_]^+^ Found: 660.33877.

*6-{4-[(4-{[(4-Cyano-1-methyl-5,6,7,8-tetrahydroisoquinolin-3-yl)oxy]methyl}-1H-1,2,3-triazol-1-yl)acetyl]piperazin-1-yl}-1,3,3-trimethyl-3,4-dihydro-1H-pyrano [3,4-c]pyridine-5-carbonitrile* (**7d**). Cream solid; yield 84%; mp 254–256 °C; IR *ν*/cm^–1^: 1669 (C=O), and 2199, 2227 (C≡N). ^1^H NMR (300 MHz, DMSO-*d_6_*/CCl_4_, 1/3): δ 1.28 (s, 6H, C(CH_3_)_2_), 1.76–1.88 (m, 4H, 6,7-CH_2_), 2.31 (s, 3H, 1′-CH_3_), 2.45 (s, 3H, 1-CH_3_), 2.57–2.63 (m, 2H, 8-CH_2_), 2.71 (s, 2H, 5′-CH_2_), 2.81–2.89 (m, 2H, 5-CH_2_), 3.58–3.76 (m, 8H, C_4_H_8_N_2_), 4.57 (s, 2H, 8′-CH_2_), 5.47 (s, 2H, NCH_2_), 5.52 (s, 2H, OCH_2_), and 7.96 (s, 1H, CH_triazole_). ^13^C NMR (75 MHz, DMSO-*d*_6_/CCl_4_, 1/3): δ 20.84, 20.95, 21.80, 21.91, 24.58, 25.70, 27.78, 37.70, 41.19, 43.94, 47.44, 47.89, 50.32, 59.16, 59.27, 69.01, 92.67, 93.21, 113.62, 115.50, 119.37, 123.71, 125.56, 147.71, 151.90, 155.34, 158.43, 159.04, 160.13, and 163.76. Anal. calcd. for C_32_H_37_N_9_O_3_: C 64.52; H 6.26; and N 21.16%. Found: 64.88; H 6.74; and N 21.41%.

*6-{4-[(4-{[(4-Cyano-1-methyl-5,6,7,8-tetrahydroisoquinolin-3-yl)oxy]methyl}-1H-1,2,3-triazol-1-yl)acetyl]piperazin-1-yl}-1-ethyl-3,3-dimethyl-3,4-dihydro-1H-pyrano [3,4-c]pyridine-5-carbonitrile* (**7e**). Cream solid; yield 71%; mp 214–216 °C; IR *ν*/cm^–1^: 1668 (C=O), and 2206, 2225 (C≡N). ^1^H NMR (300 MHz, DMSO-*d_6_*/CCl_4_, 1/3): δ 1.25 (t, *J =* 7.5 Hz, 3H, CH_2_CH_3_), 1.28 (s, 6H, C(CH_3_)_2_), 1.75–1.89 (m, 4H, 6,7-CH_2_), 2.45 (s, 3H, 1-CH_3_), 2.57 (q, *J =* 7.5 Hz, 2H, CH_2_CH_3_), 2.58–2.63 (m, 2H, 8-CH_2_), 2.72 (s, 2H, 5′-CH_2_), 2.82–2.87 (m, 2H, 5-CH_2_), 3.61–3.76 (m, 8H, C_4_H_8_N_2_), 4.60 (s, 2H, 8′-CH_2_), 5.48 (s, 2H, NCH_2_), 5.52 (s, 2H, OCH_2_), and 7.95 (s, 1H, CH_triazole_). ^13^C NMR (75 MHz, DMSO-*d_6_*/CCl_4_, 1/3): δ 10.87, 20.87, 21.84, 21.95, 24.62, 25.74, 26.44, 27.81, 37.86, 41.18, 41.21, 43.95, 47.39, 47.90, 50.34, 58.82, 59.31, 68.97, 92.70, 93.03, 113.65, 115.58, 118.75, 123.74, 125.51, 141.77, 147.85, 151.93, 158.45, 159.12, 159.38, 160.16, and 163.78. Anal. calcd. for C_33_H_39_N_9_O_3_: 65.01; H 6.45; and N 20.68%. Found: C 65.32; H 6.62; and N 20.90%.

*6-{4-[(4-{[(4-Cyano-1-methyl-5,6,7,8-tetrahydroisoquinolin-3-yl)oxy]methyl}-1H-1,2,3-triazol-1-yl)acetyl]piperazin-1-yl}-3,3-dimethyl-1-propyl-3,4-dihydro-1H-pyrano [3,4-c]pyridine-5-carbonitrile* (**7f**). Cream solid; yield 75%; mp 182–184 °C; IR *ν*/cm^–1^: 1674 (C=O) and 2220 (C≡N). ^1^H NMR (300 MHz, DMSO-*d_6_*/CCl_4_, 1/3): δ 1.00 (t, *J =* 7.5 Hz, 3H, CH_2_CH_3_), 1.28 (s, 6H, C(CH_3_)_2_), 1.67–1.89 (m, 6H, 6,7-CH_2_, CH_2_CH_3_), 2.45 (s, 3H, 1-CH_3_), 2.47–2.54 (m, 2H, CH_2_C_2_H_5_), 2.57–2.63 (m, 2H, 8-CH_2_), 2.72 (s, 2H, 5′-CH_2_), 2.80–2.91 (m, 2H, 5-CH_2_), 3.59–3.76 (m, 8H, C_4_H_8_N_2_), 4.60 (s, 2H, 8′-CH_2_), 5.47 (s, 2H, NCH_2_), 5.52 (s, 2H, OCH_2_), and 7.95 (s, 1H, CH_triazole_). ^13^C NMR (75 MHz, DMSO-*d_6_*/CCl_4_, 1/3): δ 13.49, 19.97, 20.85, 21.82, 24.59, 25.71, 27.79, 35.17, 37.85, 41.18, 43.95, 47.38, 47.94, 50.31, 58.91, 59.28, 68.93, 92.68, 93.14, 113.62, 115.53, 119.03, 123.72, 125.51, 125.54, 147.92, 151.91, 152.24, 158.43, 158.50, 158.86, 159.02, 160.14, and 163.74. Anal. calcd. for C_34_H_41_N_9_O_3_: 65.47; H 6.63; and N 20.21%. Found: C 65.82; H 6.82; and N 20.45%.

*6-{4-[(4-{[(4-Cyano-1-isopropyl-5,6,7,8-tetrahydroisoquinolin-3-yl)oxy]methyl}-1H-1,2,3-triazol-1-yl)acetyl]piperazin-1-yl}-1,3,3-trimethyl-3,4-dihydro-1H-pyrano [3,4-c]pyridine-5-carbonitrile* (**7g**). Colorless solid; yield 81%; mp 225–227 °C; IR *ν*/cm^–1^: 1668 (C=O), and 2199, 2227 (C≡N). ^1^H NMR (300 MHz, DMSO-*d_6_*/CCl_4_, 1/3): δ 1.25 (d, *J =* 6.7 Hz, 6H, CH(CH_3_)_2_), 1.28 (s, 6H, C(CH_3_)_2_), 1.79–1.86 (m, 4H, 6,7-CH_2_), 2.31 (s, 3H, 1′-CH_3_), 2.66–2.73 (m, 2H, 8-CH_2_), 2.71 (s, 2H, 5′-CH_2_), 2.84–2.89 (m, 2H, 5-CH_2_), 3.22 (sp, *J =* 6.7 Hz, 1H, CH(CH_3_)_2_), 3.58–3.75 (m, 8H, C_4_H_8_N_2_), 4.57 (s, 2H, 8′-CH_2_), 5.48 (s, 2H, NCH_2_), 5.55 (s, 2H, OCH_2_), and 7.93 (s, 1H, CH_triazole_). ^13^C NMR (75 MHz, DMSO-*d_6_*/CCl_4_, 1/3): δ 20.84, 20.90, 21.0, 21.97, 23.82, 25.74, 28.12, 30.36, 37.74, 41.24, 43.96, 47.49, 47.95, 50.34, 59.19, 69.05, 92.69, 93.26, 113.72, 115.56, 119.43, 122.26, 125.15, 141.85, 147.75, 152.55, 155.40, 159.10, 160.48, 163.80, and 166.06. Anal. calcd. for C_34_H_41_N_9_O_3_: 65.47; H 6.63; and N 20.21%. Found: C 65.86; H 6.85; and N 20.49%. ESI + MS: [C_34_H_41_N_9_O_3_Na_1_]^+^ Found: 646.32303.

*6-{4-[(4-{[(4-Cyano-1-isopropyl-5,6,7,8-tetrahydroisoquinolin-3-yl)oxy]methyl}-1H-1,2,3-triazol-1-yl)acetyl]piperazin-1-yl}-3,3-dimethyl-1-propyl-3,4-dihydro-1H-pyrano [3,4-c]pyridine-5-carbonitrile* (**7i**). Cream solid; yield 74%; mp 212–214 °C; IR *ν*/cm^–1^: 1656 (C=O) and 2215 (C≡N). ^1^H NMR (300 MHz, DMSO-*d_6_*/CCl_4_, 1/3): δ 1.00 (t, *J =* 7.3 Hz, 3H, CH_2_CH_3_), 1.25 (d, *J =* 6.7 Hz, 6H, CH(CH_3_)_2_), 1.28 (s, 6H, C(CH_3_)_2_), 1.67–1.78 (m, 2H, CH_2_CH_3_), 1.79–1.87 (m, 4H, 6,7-CH_2_), 2.48–2.52 (m, 2H, CH_2_C_2_H_5_), 2.66–2.71 (m, 2H, 8-CH_2_), 2.72 (s, 2H, 5′-CH_2_), 2.84–2.89 (m, 2H, 5-CH_2_), 3.22 (sp, *J =* 6.7 Hz, 1H, CH(CH_3_)_2_), 3.59–3.75 (m, 8H, C_4_H_8_N_2_), 4.60 (s, 2H, 8′-CH_2_), 5.47 (s, 2H, NCH_2_), 5.55 (s, 2H, OCH_2_), and 7.93 (s, 1H, CH_triazole_). ^13^C NMR (75 MHz, DMSO-*d_6_*/CCl_4_, 1/3): δ 13.51, 19.99, 20.82, 20.88, 21.96, 23.81, 25.72, 28.10, 30.34, 35.18, 37.87, 41.18, 43.94, 47.39, 47.94, 50.32, 58.94, 59.18, 68.95, 92.69, 93.15, 113.69, 115.55, 119.05, 122.22, 125.11, 141.89, 147.94, 152.52, 158.52, 159.04, 160.47, 163.73, and 166.02. Anal. calcd. for C_36_H_45_N_9_O_3_: 66.34; H 6.96; and N 19.34%. Found: C 66.67; H 7.14; and N 19.59%.

*6-(4-{[4-({[4-Cyano-1-(2-furyl)-5,6,7,8-tetrahydroisoquinolin-3-yl]oxy}methyl)-1H-1,2,3-triazol-1-yl]acetyl}piperazin-1-yl)-3,3,8-trimethyl-3,4-dihydro-1H-pyrano [3,4-c]pyridine-5-carbonitrile* (**7j**). Cream solid; yield 77%; mp 236–238 °C; IR *ν*/cm^–1^: 1662 (C=O) and 2213 (C≡N). ^1^H NMR (300 MHz, DMSO-*d_6_*/CCl_4_, 1/3): δ 1.28 (s, 6H, C(CH_3_)_2_), 1.82–1.87 (m, 4H, 6′,7′-CH_2_), 2.30 (s, 3H, CH_3_), 2.71 (s, 2H, 4-CH_2_), 2.90–2.95 (m, 2H, 8´-CH_2_), 2.99–3.04 (m, 2H, 5´-CH_2_), 3.57–3.74 (m, 8H, C_4_H_8_N_2_), 4.57 (s, 2H, 1-CH_2_), 5.47 (s, 2H, NCH_2_), 5.60 (s, 2H, OCH_2_), 6.63 (dd, *J =* 3.5,1.7 Hz, 1H, 4-CH_fur._), 7.25 (d, *J =* 3.5 Hz, 1H, 3-CH_fur._), 7.75 (d, *J =* 1.7 Hz, 1H, 5-CH_fur._), and 8.01 (s, 1H, CH_triazole_). ^13^C NMR (75 MHz, DMSO-*d_6_*/CCl_4_, 1/3): δ 20.62, 21.00, 21.93, 25.49, 25.74, 28.46, 37.72, 41.21, 43.96, 47.47, 47.48, 47.91, 50.39, 59.20, 59.45, 69.05, 93.22, 93.60, 111.60, 113.63, 114.39, 115.57, 119.41, 122.11, 125.60, 141.77, 144.19, 146.53, 147.74, 152.59, 154.16, 155.40, 159.08, 159.71, and 163.80. Anal. calcd. for C_35_H_37_N_9_O_4_: C 64.90; H 5.76; and N 19.46%. Found: C 65.27; H 5.97; and N 19.73%. ESI + MS: [C_35_H_37_N_9_O_4_Na_1_]^+^ Found: 670.28664.

*6-(4-{[4-({[4-Cyano-1-(2-furyl)-5,6,7,8-tetrahydroisoquinolin-3-yl]oxy}methyl)-1H-1,2,3-triazol-1-yl]acetyl}piperazin-1-yl)-1-ethyl-3,3-dimethyl-3,4-dihydro-1H-pyrano [3,4-c]pyridine-5-carbonitrile* (**7k**). Cream solid; yield 72%; mp 244–246 °C; IR *ν*/cm^–1^: 1672 (C=O), and 2200, 2223 (C≡N). ^1^H NMR (300 MHz, DMSO-*d_6_*/CCl_4_, 1/3): δ 1.25 (t, *J =* 7.5 Hz, 3H, CH_2_CH_3_), 1.27 (s, 6H, C(CH_3_)_2_), 1.80–1.88 (m, 4H, 6,7-CH_2_), 2.56 (q, *J =* 7.5 Hz, 2H, CH_2_CH_3_), 2.71 (s, 2H, 5′-CH_2_), 2.88–2.95 (m, 2H, 8-CH_2_), 2.98–3.04 (m, 2H, 5-CH_2_), 3.60–3.74 (m, 8H, C_4_H_8_N_2_), 4.59 (s, 2H, 8′-CH_2_), 5.48 (s, 2H, NCH_2_), 5.59 (s, 2H, OCH_2_), 6.62 (dd, *J =* 3.5,1.7 Hz, 1H, 4-CH_fur._), 7.25 (d, *J =* 3.5 Hz, 1H, 3-CH_fur._), 7.75 (d, *J =* 1.7 Hz, 1H, 5-CH_fur._), and 8.01 (s, 1H, CH_triazole_). ^13^C NMR (75 MHz, DMSO-*d_6_*/CCl_4_, 1/3): δ 10.86, 20.60, 21.91, 25.47, 25.72, 26.43, 28.44, 37.83, 41.19, 43.95, 47.36, 47.87, 50.36, 58.81, 59.44, 68.97, 92.98, 93.59, 111.58, 113.61, 114.37, 115.60, 118.75, 122.10, 125.59, 141.79, 144.16, 146.52, 147.83, 152.58, 154.14, 159.09, 159.38, 159.71, and 163.77. Anal. calcd. for C_36_H_39_N_9_O_4_: C 65.34; H 5.94; and N 19.05%. Found: C 65.66; H 6.11; and N 19.28%.

*6-(4-{[4-({[4-Cyano-1-(2-furyl)-5,6,7,8-tetrahydroisoquinolin-3-yl]oxy}methyl)-1H-1,2,3-triazol-1-yl]acetyl}piperazin-1-yl)-3,3-dimethyl-8-propyl-3,4-dihydro-1H-pyrano [3,4-c]pyridine-5-carbonitrile* (**7l**). Brown solid; yield 79%; mp 144–146 °C; IR *ν*/cm^–1^: 1654 (C=O) and 2215 (C≡N). ^1^H NMR (300 MHz, DMSO-*d_6_*/CCl_4_, 1/3): δ 0.99 (t, *J =* 7.4 Hz, 3H, CH_2_CH_3_), 1.27 (s, 6H, C(CH_3_)_2_), 1.66–1.80 (m, 2H, CH_2_CH_3_), 1.81–1.88 (m, 4H, 6′,7′-CH_2_), 2.43–2.54 (m, 2H, 8´-CH_2_), 2.72 (s, 2H, 4-CH_2_), 2.88–2.95 (m, 2H, CH_2_C_2_H_5_), 2.97–3.05 (m, 2H, 5´-CH_2_), 3.58–3.74 (m, 8H, C_4_H_8_N_2_), 4.60 (s, 2H, 1-CH_2_), 5.47 (s, 2H, NCH_2_), 5.60 (s, 2H, OCH_2_), 6.62 (dd, *J =* 3.5,1.7 Hz, 1H, 4-CH_fur._), 7.25 (d, *J =* 3.5 Hz, 1H, 3-CH_fur._), 7.74 (d, *J =* 1.7 Hz, 1H, 5-CH_fur._), and 8.01 (s, 1H, CH_triazole_). ^13^C NMR (75 MHz, DMSO-*d_6_*/CCl_4_, 1/3): δ 13.53, 20.01, 20.63, 21.94, 25.49, 25.73, 28.46, 35.20, 37.87, 41.21, 43.96, 47.41, 47.93, 50.37, 58.94, 59.45, 68.97, 93.14, 93.61, 111.59, 113.62, 114.38, 115.58, 119.06, 122.12, 125.58, 144.17, 146.55, 147.95, 152.60, 154.16, 158.55, 159.04, and 163.78. Anal. calcd. for C_37_H_41_N_9_O_4_: C 65.76; H 6.12; and N 18.65%. Found: C 66.12; H 6.32; and N 18.90%. ESI + MS: [C_37_H_41_N_9_O_4_Na_1_]^+^ Found: 698.31793.

*6-{4-[(4-{[(5-Cyano-3,3,8-trimethyl-3,4-dihydro-1H-pyrano [3,4-c]pyridin-6-yl)oxy]methyl}-1H-1,2,3-triazol-1-yl)acetyl]piperazin-1-yl}-3,3,8-trimethyl-3,4-dihydro-1H-pyrano [3,4-c]pyridine-5-carbonitrile* (**7m**). Colorless solid; yield 85%; mp 224–226 °C; IR *ν*/cm^–1^: 1673 (C=O), and 2210, 2228 (C≡N). ^1^H NMR (300 MHz, DMSO-*d_6_*/CCl_4_, 1/3): δ 1.28 (s, 12H, 2C(CH_3_)_2_), 2.30 (s, 3H, CH_3_), 2.40 (s, 3H, CH_3_), 2.71 (s, 2H, 4-CH_2_), 2.75 (s, 2H, 4′-CH_2_), 3.57–3.75 (m, 8H, C_4_H_8_N_2_), 4.57 (s, 2H, 1-CH_2_), 4.61 (s, 2H, 1′-CH_2_), 5.48 (s, 2H, NCH_2_), 5.54 (s, 2H, OCH_2_), and 7.98 (s, 1H, CH_triazole_). ^13^C NMR (75 MHz, DMSO-*d_6_*/CCl_4_, 1/3): δ 20.81, 21.01, 25.68, 25.74, 37.57, 37.73, 41.22, 43.98, 47.48, 47.94, 50.37, 59.19, 59.52, 69.01, 69.05, 92.96, 93.26, 113.29, 115.57, 119.43, 121.00, 125.70, 141.52, 147.75, 148.74, 155.31, 155.40, 159.09, 160.84, and 163.82. Anal. calcd. for C_33_H_39_N_9_O_4_: C 63.34; H 6.28; and N 20.15%. Found: C 63.73; H 6.52; and N 20.43%. ESI + MS: [C_33_H_39_N_9_O_4_Na_1_]^+^ Found: 648.30223.

*6-{4-[(4-{[(5-Cyano-3,3,8-trimethyl-3,4-dihydro-1H-pyrano [3,4-c]pyridin-6-yl)oxy]methyl}-1H-1,2,3-triazol-1-yl)acetyl]piperazin-1-yl}-1-ethyl-3,3-dimethyl-3,4-dihydro-1H-pyrano [3,4-c]pyridine-5-carbonitrile* (**7n**). Colorless solid; yield 87%; mp 222–224 °C; IR *ν*/cm^–1^: 1651 (C=O), and 2205, 2225 (C≡N). ^1^H NMR (300 MHz, DMSO-*d_6_*/CCl_4_, 1/3): δ 1.25 (t, *J =* 7.5 Hz, 3H, CH_2_CH_3_), 1.27 (s, 6H, C(CH_3_)_2_), 1.28 (s, 6H, C(CH_3_)_2_), 2.40 (s, 3H, CH_3_), 2.57 (q, *J =* 7.5 Hz, 2H, CH_2_CH_3_), 2.72 (s, 2H, 4-CH_2_), 2.75 (s, 2H, 4′-CH_2_), 3.61–3.75 (m, 8H, C_4_H_8_N_2_), 4.60 (s, 2H, 1-CH_2_), 4.61 (s, 2H, 1′-CH_2_), 5.48 (s, 2H, NCH_2_), 5.55 (s, 2H, OCH_2_), and 7.97 (s, 1H, CH_triazole_). ^13^C NMR (75 MHz, DMSO-*d_6_*/CCl_4_, 1/3): δ 10.88, 20.79, 25.68, 25.75, 26.45, 37.57, 37.86, 41.20, 43.92, 43.96, 47.41, 47.91, 50.35, 58.83, 59.18, 59.53, 68.99, 92.98, 93.03, 113.26, 115.61, 118.77, 120.99, 125.68, 141.54, 147.86, 148.73, 155.30, 159.12, 159.39, 160.84, and 163.79. Anal. calcd. for C_34_H_41_N_9_O_4_: C 63.83; H 6.46; and N 19.70%. Found: C 64.16; H 6.65; and N 19.94%.

*6-{4-[(4-{[(5-Cyano-8-ethyl-3,3-dimethyl-3,4-dihydro-1H-pyrano [3,4-c]pyridin-6-yl)oxy]methyl}-1H-1,2,3-triazol-1-yl)acetyl]piperazin-1-yl}-1,3,3-trimethyl-3,4-dihydro-1H-pyrano [3,4-c]pyridine-5-carbonitrile* (**7p**). Colorless solid; yield 78%; mp 239–241 °C; IR *ν*/cm^–1^: 1671 (C=O), and 2211, 2224 (C≡N). ^1^H NMR (300 MHz, DMSO-*d_6_*/CCl_4_, 1/3): δ 1.27 (s, 12H, 2C(CH_3_)_2_), 1.32 (t, *J =* 7.5 Hz, 3H, CH_2_CH_3_), 2.31 (s, 3H, CH_3_), 2.65 (q, *J =* 7.5 Hz, 2H, CH_2_CH_3_), 2.71 (s, 2H, 4-CH_2_), 2.75 (s, 2H, 4′-CH_2_), 3.58–3.75 (m, 8H, C_4_H_8_N_2_), 4.57 (s, 2H, 1-CH_2_), 4.64 (s, 2H, 1′-CH_2_), 5.48 (s, 2H, NCH_2_), 5.57 (s, 2H, OCH_2_), and 7.96 (s, 1H, CH_triazole_). ^13^C NMR (75 MHz, DMSO-*d_6_*/CCl_4_, 1/3): δ 11.04, 21.0, 25.68, 25.74, 26.36, 37.66, 37.71, 41.21, 43.95, 47.47, 47.50, 47.92, 50.36, 58.82, 59.42, 59.45, 69.05, 92.87, 93.25, 113.32, 115.56, 119.42, 120.37, 125.48, 141.60, 147.76, 148.87, 155.40, 159.08, 159.51, 161.05, and 163.80. Anal. calcd. for C_34_H_41_N_9_O_4_: C 63.83; H 6.46; and N 19.70%. Found: C 64.21; H 6.68; and N 19.97%.

**6**-*{4-[(4-{[(5-Cyano-8-ethyl-3,3-dimethyl-3,4-dihydro-1H-pyrano [3,4-c]pyridin-6-yl)oxy]methyl}-1H-1,2,3-triazol-1-yl)acetyl]piperazin-1-yl}-8-ethyl-3,3-dimethyl-3,4-dihydro-1H-pyrano [3,4-c]pyridine-5-carbonitrile* (**7q**). Colorless solid; yield 83%; mp 217–219 °C; IR *ν*/cm^–1^: 1676 (C=O), and 2214, 2234 (C≡N). ^1^H NMR (300 MHz, DMSO-*d_6_*/CCl_4_, 1/3): δ 1.25 (t, *J =* 7.4 Hz, 3H, CH_2_CH_3_), 1.27 (s, 12H, 2C(CH_3_)_2_), 1.32 (t, *J =* 7.4 Hz, 3H, CH_2_CH_3_), 2.57 (q, *J =* 7.4 Hz, 2H, CH_2_CH_3_), 2.65 (q, *J =* 7.4 Hz, 2H, CH_2_CH_3_), 2.72 (s, 2H, 4-CH_2_), 2.76 (s, 2H, 4′-CH_2_), 3.61–3.75 (m, 8H, C_4_H_8_N_2_), 4.60 (s, 2H, 1-CH_2_), 4.64 (s, 2H, 1′-CH_2_), 5.48 (s, 2H, NCH_2_), 5.57 (s, 2H, OCH_2_), and 7.96 (s, 1H, CH_triazole_). ^13^C NMR (75 MHz, DMSO-*d_6_*/CCl_4_, 1/3): δ 10.88, 11.04, 25.68, 25.75, 26.36, 26.44, 37.70, 37.86, 41.19, 43.95, 47.40, 47.92, 50.35, 58.82, 59.45, 68.95, 68.99, 92.87, 93.03, 113.32, 115.61, 118.78, 120.38, 125.48, 141.60, 147.86, 148.87, 159.13, 159.40, 159.51, 161.06, and 163.79. Anal. calcd. for C_35_H_43_N_9_O_4_: C 64.30; H 6.63; and N 19.28%. Found: C 64.62; H 6.80; and N 19.51%. ESI + MS: [C_35_H_43_N_9_O_4_Na_1_]^+^ Found: 676.33365.

*6-[(1-{2-[4-(5-Cyano-3,3-dimethyl-8-propyl-3,4-dihydro-1H-pyrano [3,4-c]pyridin-6-yl)piperazin-1-yl]-2-oxoethyl}-1H-1,2,3-triazol-4-yl)methoxy]-8-ethyl-3,3-dimethyl-3,4-dihydro-1H-pyrano [3,4-c]pyridine-5-carbonitrile* (**7r**). Colorless solid; yield 76%; mp 206–208 °C; IR *ν*/cm^–1^: 1679 (C=O), and 2211, 2229 (C≡N). ^1^H NMR (300 MHz, DMSO-*d_6_*/CCl_4_, 1/3): δ1.00 (t, *J =* 7.3 Hz, 3H, CH_2_CH_2_CH_3_), 1.27 (s, 12H, 2C(CH_3_)_2_), 1.32 (t, *J =* 7.4 Hz, 3H, CH_2_CH_3_), 1.67–1.80 (m, 2H, CH_2_CH_2_CH_3_), 2.48–2.54 (m, 2H, CH_2_C_2_H_5_), 2.65 (q, *J =* 7.4 Hz, 2H, CH_2_CH_3_), 2.72 (s, 2H, 4-CH_2_), 2.76 (s, 2H, 4′-CH_2_), 3.60–3.76 (m, 8H, C_4_H_8_N_2_), 4.60 (s, 2H, 1-CH_2_), 4.64 (s, 2H, 1′-CH_2_), 5.48 (s, 2H, NCH_2_), 5.57 (s, 2H, OCH_2_), and 7.96 (s, 1H, CH_triazole_). ^13^C NMR (75 MHz, DMSO-*d_6_*/CCl_4_, 1/3): δ 11.04, 13.53, 19.99, 25.68, 25.73, 26.36, 35.20, 37.70, 37.87, 41.20, 43.96, 47.40, 47.95, 47.98, 50.35, 58.81, 58.94, 59.45, 68.94, 68.97, 92.88, 93.17, 113.29, 115.57, 119.08, 120.36, 125.45, 141.59, 147.95, 148.86, 158.53, 159.04, 159.49, 161.04, and 163.77. Anal. calcd. for C_36_H_45_N_9_O_4_: C 64.75; H 6.79; and N 18.88%. Found: C 65.10; H 6.98; and N 19.14%. ESI + MS: [C_36_H_45_N_9_O_4_Na_1_]^+^ Found: 690.34927.

*6-{4-[(4-{[(5-Cyano-3,3-dimethyl-8-propyl-3,4-dihydro-1H-pyrano [3,4-c]pyridin-6-yl)oxy]methyl}-1H-1,2,3-triazol-1-yl)acetyl]piperazin-1-yl}-3,3,8-trimethyl-3,4-dihydro-1H-pyrano [3,4-c]pyridine-5-carbonitrile* (**7s**). Colorless solid; yield 79%; mp 192–194 °C; IR *ν*/cm^–1^: 1672 (C=O) and 2213 (C≡N). ^1^H NMR (300 MHz, DMSO-*d_6_*/CCl_4_, 1/3): δ1.03 (t, *J =* 7.4 Hz, 3H, CH_2_CH_3_), 1.27 (s, 6H, C(CH_3_)_2_), 1.28 (s, 6H, C(CH_3_)_2_), 1.74–1.87 (m, 2H, CH_2_CH_3_), 2.31 (s, 3H, CH_3_), 2.56–2.62 (m, 2H, CH_2_C_2_H_5_), 2.71 (s, 2H, 4-CH_2_), 2.76 (s, 2H, 4′-CH_2_), 3.58–3.75 (m, 8H, C_4_H_8_N_2_), 4.57 (s, 2H, 1-CH_2_), 4.65 (s, 2H, 1′-CH_2_), 5.48 (s, 2H, NCH_2_), 5.55 (s, 2H, OCH_2_), and 7.96 (s, 1H, CH_triazole_). ^13^C NMR (75 MHz, DMSO-*d_6_*/CCl_4_, 1/3): δ 13.53, 20.17, 21.00, 25.68, 25.74, 35.07, 37.73, 41.22, 43.98, 47.48, 47.94, 50.37, 58.95, 59.20, 59.50, 68.93, 69.05, 92.94, 93.26, 113.31, 115.57, 119.44, 120.68, 125.44, 141.56, 147.76, 148.96, 155.40, 158.65, 159.09, 160.94, and 163.80. Anal. calcd. for C_35_H_43_N_9_O_4_: C 64.30; H 6.63; and N 19.28%. Found: C 64.70; H 6.86; and N 19.57%. ESI + MS: [C_35_H_43_N_9_O_4_Na_1_]^+^ Found: 676.33365.

*6-{4-[(4-{[(5-Cyano-3,3-dimethyl-8-propyl-3,4-dihydro-1H-pyrano [3,4-c]pyridin-6-yl)oxy]methyl}-1H-1,2,3-triazol-1-yl)acetyl]piperazin-1-yl}-1-ethyl-3,3-dimethyl-3,4-dihydro-1H-pyrano [3,4-c]pyridine-5-carbonitrile* (**7t**). Colorless solid; yield 81%; mp 170–172 °C; IR *ν*/cm^–1^: 1675 (C=O) and 2222 (C≡N). ^1^H NMR (300 MHz, DMSO-*d_6_*/CCl_4_, 1/3): δ 1.03 (t, *J =* 7.4 Hz, 3H, C_2_H_4_CH_3_), 1.24 (t, *J =* 7.4 Hz, 3H, CH_2_CH_3_), 1.27 (s, 6H, C(CH_3_)_2_), 1.28 (s, 6H, C(CH_3_)_2_), 1.74–1.87 (m, 2H, CH_2_CH_2_CH_3_), 2.53–2.62 (m, 4H, C_2_H_5_CH_2_, CH_2_CH_3_), 2.72 (s, 2H, 4-CH_2_), 2.76 (s, 2H, 4’-CH_2_), 3.61–3.76 (m, 8H, C_4_H_8_N_2_), 4.60 (s, 2H, 1-CH_2_), 4.64 (s, 2H, 1′-CH_2_), 5.48 (s, 2H, NCH_2_), 5.55 (s, 2H, OCH_2_), and 7.95 (s, 1H, CH_triazole_). ^13^C NMR (75 MHz, DMSO-*d_6_*/CCl_4_, 1/3): δ 10.88, 13.52, 20.17, 25.59, 25.67, 25.75, 26.45, 35.07, 37.73, 37.86, 41.19, 43.98, 47.39, 47.92, 50.35, 58.82, 58.95, 59.52, 68.98, 92.96, 93.05, 113.29, 115.59, 118.78, 120.67, 125.41, 141.57, 147.86, 148.96, 158.63, 159.13, 159.38, 160.94, and 163.78. Anal. calcd. for C_36_H_45_N_9_O_4_: C 64.75; H 6.79; and N 18.88%. Found: C 65.13; H 6.99; and N 19.15%.

*6-{4-[(4-{[(5-Cyano-3,3-dimethyl-8-propyl-3,4-dihydro-1H-pyrano [3,4-c]pyridin-6-yl)oxy]methyl}-1H-1,2,3-triazol-1-yl)acetyl]piperazin-1-yl}-3,3-dimethyl-8-propyl-3,4-dihydro-1H-pyrano [3,4-c]pyridine-5-carbonitrile* (**7u**). Colorless solid; yield 88%; mp 211–213 °C; IR *ν*/cm^–1^: 1662 (C=O), and 2209, 2231 (C≡N). ^1^H NMR (300 MHz, DMSO-*d_6_*/CCl_4_, 1/3): δ 1.00 (t, *J =* 7.3 Hz, 3H, CH_2_CH_3_), 1.03 (t, *J =* 7.3 Hz, 3H, CH_2_CH_3_), 1.28 (s, 12H, 2C(CH_3_)_2_), 1.67–1.87 (m, 4H, 2CH_2_CH_3_), 2.53–2.62 (m, 4H, 2CH_2_C_2_H_5_), 2.72 (s, 2H, 4-CH_2_), 2.76 (s, 2H, 4′-CH_2_), 3.60–3.76 (m, 8H, C_4_H_8_N_2_), 4.60 (s, 2H, 1-CH_2_), 4.65 (s, 2H, 1′-CH_2_), 5.48 (s, 2H, NCH_2_), 5.55 (s, 2H, OCH_2_), and 7.96 (s, 1H, CH_triazole_). ^13^C NMR (75 MHz, DMSO-*d_6_*/CCl_4_, 1/3): δ 13.53, 20.01, 20.17, 25.68, 25.71, 25.74, 35.07, 35.20, 37.73, 37.88, 41.19, 43.97, 47.41, 47.98, 50.35, 58.95, 59.51, 68.93, 68.97, 92.94, 93.17, 113.30, 115.58, 119.08, 120.67, 125.43, 141.57, 147.96, 148.96, 158.55, 158.64, 159.06, 160.94, and 163.78. Anal. calcd. for C_37_H_47_N_9_O_4_: C 65.18; H 6.95; and N 18.49%. Found: C 65.49; H 7.12; and N 18.72%. ESI + MS: [C_37_H_47_N_9_O_4_Na_1_]^+^ Found: 704.36494.

### 3.2. Biological Evaluation

Compounds were studied for their possible neurotropic activities (anticonvulsant, sedative, and anti-anxiety activities) as well as for side effects on 450 white mice of both sexes weighing 18–24 g and 50 male rats of the Wistar line weighing 120–140 g. All groups of animals were maintained at 25 ± 2 °C in the same room on a common food ration. As reference compounds, the known antiepileptic drug ethosuximide and tranquilizer diazepam were used. All the biological experiments were carried out in full compliance with the European Convention for the Protection of Vertebrates. All animal procedures were performed in accordance with the Guidelines for Care and Use of Laboratory Animals of Strasbourg (France; ETS No 123, Strasbourg, 03/18/1986).

#### 3.2.1. Evaluation of the Anticonvulsant Activity of the Synthesized Compounds

The anticonvulsant effect of the newly synthesized compounds was investigated by tests comprising pentylenetetrazole, thiosemicarbazide convulsions, and maximal electroshock (MES). Outbred mice (weight 18–22 g) were used for the study. In the case of convulsions induced by PTZ, it was injected subcutaneously at 90 mg/kg, which induced convulsions in 95% of animals (CD 95%). Each animal is placed into an individual plastic cage for observation lasting 1 h. Seizures and clonic convulsions were recorded. Substances were administered intraperitoneally (i.p.) at doses of 10–200 mg/kg in suspension with carboxymethylcellulose and Tween-80 45 min before administration of PTZ and applying electrical stimulation. The control animals were administered as an emulsifier. Every dose of each test compound was studied in six animals.

The MES test is used as an animal model for the generalized tonic seizures of epilepsy. The parameters of MES were: 50 mA, duration of 0.2 s, and an oscillation frequency of 50 imp/s. The anticonvulsant properties of compounds were assessed by their prevention of the tonic-extensor phase of convulsions.

Thiosemicarbazide, an antimetabolite of GABA inhibitor (glutamic acid decarboxylase) in the brain, is administered subcutaneously to mice at a dose of 18 mg/kg as a 0.5% solution, which causes clonic convulsions in animals. Anti-thiosemicarbazide activity was evaluated based on the latency time of the onset of seizures. Compounds were administered intraperitoneally at doses of 100 mg/kg in suspension with carboxymethylcellulose and Tween-80 45 min before administration of thiosemicarbazide.

The comparative drug ethosuximide was administered in doses of 200 mg/kg and diazepam in doses of 2 mg/kg.

#### 3.2.2. Evaluation of the Psychotropic Properties of the Synthesized Compounds

Psychotropic properties of selected compounds were studied by the following tests: “open field”, “elevated plus maze” (EPM), “forced swimming”, and “test for learning and memory”.

Open field test. The research-motor behavior of rats was studied on a modified “open field” model. For this purpose, an installation was used, the bottom of which is divided into squares with holes (cells). Experiments were performed in the daytime with natural light. Within 5 min of the experiment, the indicators of sedative and activating behavior were determined, namely the number of horizontal movements, standing on the hind legs (vertical movements), and sniffing of the cells. The number of animals on this model was eight for each compound, control, and reference drug. The studied compounds were administered to rats at the most effective dose of 50 mg/kg intraperitoneally as a suspension with methylcarboxycellulose and Tween-80.

Elevated plus maze—EPM test. Anti-anxiety and sedative effects were studied on mouse a model of the “elevated plus maze”. The labyrinth is a cruciform machine raised above the floor, with a pair of open and closed sleeves opposed to each other. Normal animals prefer to spend most of their time in the closed (dark) sleeves of the labyrinth. The anxiolytic effect of the compounds was estimated by the increase in the number of entries into open (light) sleeves and the time spent in them without increasing the total motor activity. This records the time spent in the closed sleeve and the number of attempts to enter the installation center. In the above model, the test compounds and the reference drug were injected intraperitoneally before the experiments. The control animals were administered as an emulsifier. Results were processed statistically (*p* < 0.05).

Forced swimming test. To assess “despair and depression”, the “compelling swimming” model was used. Experimental animals were forced to swim in a glass container (height 22 cm, diameter 14 cm), filled 1/3 with water. Intact mice swim very actively, but soon they will be forced to immobilize. The latent period of immobilization, the total duration of active swimming and immobilization are fixed at 6 min (the experiments were conducted under natural light).

Test for learning and memory. The antiamnesic properties of the compounds were studied using an electroshock amnesia model modified by J. Bures, O. Buresova [43]. In rats, a conditioned passive avoidance reaction (CRPA) to a dark-light setting was developed, while the training of the animals with CRPA was recorded. The residence times in the light and dark compartments were recorded for 5 min (300 s). Then, in the dark compartment, the animal received a single current shock (0.4 mA) through the electrode floor (training). To obtain amnesia, a maximum electroconvulsive seizure is induced in animals (conducting a maximum electric current of 50 Hz for 0.2 s through corneal electrodes). Conducting an electric shock immediately after training causes the erasure of the memory trace. The test for reproduction is carried out 24 h after training—after the development of CRPA. During this period, the control animals forget the training and prefer to be in the dark part of the chamber. An increase in the time spent by rats in the light compartment compared to the control on the second day indicates the presence of antiamnestic properties of the compounds.

#### 3.2.3. Evaluation of Coordination of Movement in the Rotating Rod Test

Adverse neurotoxic (muscle relaxant) effects of compounds were studied at doses of 50 to 500 mg/kg when administered intraperitoneally, as well as reference drugs in effect at anticonvulsant doses. Myorelaxation was investigated with the test of a “rotating rod” in mice. To this end, mice were planted on a metal rod with a corrugated rubber coating, which rotated at a speed of 5 revolutions per minute. The number of animals that cnot stay on it for 2 min was determined. To determine the ED50, neurotoxic TD50, and LD50, the statistical method of penetration by Litchfield and Wilcoxon was used. The acute toxicity (LD50) was determined by calculating the number of dead animals after 24 h of exposure at doses of 100–1020 mg/kg.

### 3.3. Docking Studies

Docking studies were performed using AutoDock 4 (ver. 4.2.6) into the 3D structures of the GABA_A_ receptor (PDB code: 4COF), SERT transporter (PDB code: 3F3A), and 5-HT_1A_ receptor (PDB code: 3NYA), retrieved from the protein data bank (PDB). For the final preparation of both ligands and proteins, Wizard of AutoDock Tools 1.5.6 was used.

All molecules were sketched in the chemdraw12.0 program. The geometry of built compounds was optimized using the molecular mechanical force fields 94 (MMFF94) energy via the program LigandScout (ver. 4.4.5); partial charges were also calculated; conformers of each ligand were generated; and the one with the best conformation was maintained and saved as a mol2 file that was passed to ADT for pdbqt file preparation. There, polar hydrogen was added to each structure, followed by computing the Gasteiger and Kollman charges and the torsions. The region of interest, used by Autodock4 for docking runs and by Autogrid4 for affinity grid map preparation, was defined in such a way as to comprise the whole catalytic binding site using a grid size of 110 × 110 × 110 xyz points with a grid spacing of 0.375 Å. The grid center was calculated at x = –20.558, y = –19.574 and z = 127.994 for GABA_A_ receptor, at x = –19.7478, y = 22.417, and z = –14.3006 for the SERT transporter and at x = –8.207, y = 9.305, and z = –48.61 for the 5-HT_1A_ receptor. For the simulation, default values for quaternation, translation, and torsion steps were applied. The Lamarckian Genetic Algorithm with default parameters was applied for minimization. The number of docking runs was 100. After docking, the 100 solutions were clustered into groups with RMS lower than 1.0 Ε. The clusters were ranked by the lowest energy representative of each cluster. Upon completion of docking, the best poses were screened by examination of binding energy (Δ*G*_binding_, kcal/mol) and cluster number. In order to describe the ligand-binding pocket interactions, the top-ranked binding mode found by AutoDock in complex with the binding pocket of an enzyme was selected. The Accelrys Discovery Studio 2020 Client (Dassault Systèmes presence, Wlatham, MA, USA) [54,55] and LigandScout (ver. 4.4.5) were used for the graphical representations of all ligand-protein complexes.

### 3.4. Drug Likeness

Absorption, distribution profile, and drug-likeness model score of compounds were predicted using the Molsoft software [56] Molsoft LLC., San Diego, CA, USA, and the ADMET structure-activity relationship server, admetSAR version 2.0, Shanghai, China [57] via ChemAxon’s Marvin JS structure drawing tool.

## 4. Conclusions

In summary, new original hybrids were synthesized based on the biologically active bicyclic pyridine derivatives via a Cu-catalyzed azide-alkyne 1,3-dipolar cycloaddition. The studies have shown that the alkylation of 3(6)-hydroxy derivatives of bicyclic pyridines leads to the formation of *O*-alkylated derivatives **2** in high yields, and only traces of *N*-alkylated derivatives are observed.

The anticonvulsant activity, combined with some psychotropic properties, of the new hybrids was evaluated. The biological assays evidenced that 15 of the studied compounds exhibited a high anticonvulsant activity due to their antagonism of pentylenetetrazole, which was superior to ethosuximide but inferior to diazepam. Compounds **7a**, **d**, **g**, **j**, **m**, and **o** displayed the highest therapeutic indexes among the tested ones. The compounds at 100 mg/kg dose increased latency of thiosemicarbazide seizures to 1.5–4.37 times compared to the control and reference drug diazepam. The findings suggest some GABA-ergic activity involvement in the mechanism of action of the substances. The toxicity of compounds is low and does not induce muscle relaxation at the studied doses. According to the study of psychotropic activity, it was found that the selected compounds had activating behavior, and anxiolytic effects on the models of “open field” (the data obtained indicated the anxiolytic (anti-anxiety) activity of the compounds, especially pronounced in the compounds **7a**, **d**, **g**, **m**, **j**, and **o**) and EPM (after the administration of all compounds, the experimental animals, in contrast to the control animals, enter the open arms and stay there from 6.6 (compound **7g**) to 66.4 (compound **7m**) sec. On the “forced swimming” model, some of the selected compounds **(7a**, **c**, **g**, **h**, **j**, **m**, **o**, and **q**) increased the active swimming time and the latent period of first immobilization, exhibiting some antidepressant effect similarly to diazepam. On the model of electroshock retrograde amnesia—conditioned response of passive avoidance (CRPA), the administration of compounds at a dose of 50 mg/kg per day, except for compound **7j**, caused some increase in the time of reproduction of the reflex in animals. Statistically, these values are significantly different from the control values and indicate the antiamnesic effect of the compounds.

Molecular docking studies were in agreement with experimental data.

The prediction of drug likeness revealed that the drug-likeness score was found to be from 0.33 to 0.83, indicating the possibility for these compounds to be treated as drug candidates. Furthermore, according to prediction, all the tested molecules were able to pass the blood-brain barrier (BBB). The admetSAR server revealed that all compounds are both good P-glycoprotein substrates (65–73%) and inhibitors (74–78%).

Thus, compounds **7a**, **d**, **g**, **j**, and **m**, in particular, show high anticonvulsant activity by the corazol antagonism test and simultaneously express psychotropic properties. These five most active substances contain a pyrano [3,4-*c*]pyridine cycle with a methyl group in the pyridine ring of their structure.

The obtained results demonstrated that the compounds can be effective in various types of human epilepsy associated with mental disorders.

## Data Availability

Not applicable.

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
