# Peer review of "New Bicyclic Pyridine-Based Hybrids Linked to the 1,2,3-Triazole Unit: Synthesis via Click Reaction and Evaluation of Neurotropic Activity and Molecular Docking"

_molecules, 2023, doi:10.3390/molecules28030921_

Round 1

Reviewer 1 Report

New CNS active agents have been prepared and evaluated for activity as anti-convulsant agents as well as other neurological effects.  The rationale for the study is reasonable and descriptions of experimental procedures are adequate.  New compounds appear to have been appropriately characterized.  Overall the work is suitable for publication after relatively minor revisions as indicated below.

1.  More careful proofreading for English language and formatting needs to be performed.  For example, compound numbers should appear in bold face type throughout.  Spelling of certain compounds varied in the manusciript (e.g., pentylenetetrazole and pentilenetetrazole). 

2. The data gathered from mice/rat experiments is presented solely in the form of numerical tables.  This makes it difficult to readily discern differences among the compounds tested.  The data should be presented in additional formats (such as bar graphs, 3D graphs, etc) in order to more clearly display the results and identify the most active compounds for the effect being examined.

3.  No pharmacokinetic data is provided.  What is the bioavailability of these compounds, particularly with respect to localization in the brain?  Biostability?  These issues should be accounted for.      

Reviewer 2 Report

In this manuscript by Sirakanyan et al, the authors carried out click reactions to synthesize new bicyclic pyridine-based hybrids linked to 1,2,3-triazole, then evaluated their neurotropic activity by biological assays, and conducted molecular docking studies in the end. The manuscript has to be improved according to the following points before it can be further considered for publication.

1. Molecular docking to the GABAA receptor:

- There are currently 25 experimental structures of the GABAA receptor on Protein Data Bank, according to the UniProt ID P28472. The authors should briefly explain why they chose the PDB ID 4COF as template for this docking study.

- The authors should briefly describe how they did the re-docking step. For example: Was the conformer of the co-crystal ligand kept as is before re-docking? Were extra preparation steps (energy minimization, conformer generation, etc) carried out before the ligand was re-docked?

- The authors should specify the bond lengths and bond angles for all hydrogen bonds of compound 7j.

2. Molecular docking to the SERT transporter:

- Here the authors used a crystal structure of LeuT, a prokaryotic homologue of SERT, to carry out docking. Has this practice been observed in the literature? Is there any published study that used LeuT for structure-based drug design on SERT, with validated results? Although the two proteins are prokaryotic homologues, there may be some differences in terms of structural conformations, notably in the binding site, that may affect structure-based modeling. More valid reasons for this practice should be given here.

- On Protein Data Bank, there are now 63 experimental structures of LeuT (UniProt ID O67854). The authors should briefly explain why they chose the PDB ID 3F3A here.

- Had the authors done re-docking for the co-crystal ligand of LeuT before they docked their compounds? More details should be provided.

- The authors should specify the bond angles for all hydrogen bonds of compound 7a.

- Line 331: the authors stated that the bond length of one of the hydrogen bonds was 5.58 Angstroms. In fact, this distance is way too large for a hydrogen bond! According to O. Korb et al, J Chem Inf Model 2009, 49, 84-96 (https://pubmed.ncbi.nlm.nih.gov/19125657/), hydrogen bond lengths are between 2.3 and 3.4 Angstroms. A distance outside this range is deemed too far for the formation of a hydrogen bond. The authors have to revise this detail before re-submission.

3. Molecular docking to the 5-HT1A receptor:

- There are now 38 experimental structures of 5-HT1A on Protein Data Bank (UniProt ID P07550). The authors should briefly explain why they chose the PDB ID 3NYA.

- The authors should briefly describe how they did the re-docking step (see comments above).

- The authors should specify the bond lengths and bond angles for all hydrogen bonds of compound 7a.

4. Docking studies - Materials and Methods (page 24):

- The authors have to specify which program(s) (along with versions) they used for the 3D conversion of molecules (after 2D sketching with ChemDraw), and for the addition of hydrogens into their 3D structures.

- The authors have to explain how the grid boxes were designed: the center coordinates and the sizes of the three dimensions were determined based on which factors?

- The authors should state that only one docked pose was retained for each docked ligand. In this case, they have to specify the criterion/criteria for this selection.

5. Minor modifications:

There are also many minor modifications that have to be made. For example:

- Line 27: it should be "... compounds were evaluated", not "was"

- Line 30: it should be "... their structures" ("structures" in plural)

- Line 63: it should be "an IC50 twice as low as that of the clinically used ..."

- Scheme 3: it should be "hybrids" below the compound 7, not "hydrids"

- Lines 329, 331: the symbol for "Angstrom" has to be "Å", not "A"

- The authors should indicate the bond lengths for the hydrogen bonds in their Figures 3, 4, 5, 7.

I am looking forward to the revised version of this manuscript, and willing to reconsider it for publication in Molecules.

Round 2

Reviewer 2 Report

In this revised version, the authors addressed my comments on the previous version of their manuscript. I therefore recommend accepting this paper for publication in Molecules.